# State of the Art MR Imaging for Lung Cancer TNM Stage Evaluation

**DOI:** 10.3390/cancers15030950

**Published:** 2023-02-02

**Authors:** Yoshiharu Ohno, Yoshiyuki Ozawa, Hisanobu Koyama, Takeshi Yoshikawa, Daisuke Takenaka, Hiroyuki Nagata, Takahiro Ueda, Hirotaka Ikeda, Hiroshi Toyama

**Affiliations:** 1Department of Radiology, Fujita Health University School of Medicine, Toyoake 470-1192, Japan; 2Joint Research Laboratory of Advanced Medical Imaging, Fujita Health University School of Medicine, Toyoake 470-1192, Japan; 3Department of Radiology, Osaka Police Hospital, Osaka 543-0035, Japan; 4Department of Diagnostic Radiology, Hyogo Cancer Center, Akashi 673-0021, Japan

**Keywords:** lung, MRI, PET/MRI, lung cancer, TNM staging

## Abstract

**Simple Summary:**

Magnetic resonance (MR) imaging had limited clinical indications in patients with thoracic malignancies in the last a few decades. However, technical advances in MR system, sequence, receiver coils with parallel imaging capability and reconstruction methods and clinical protocol adjustment including gadolinium contrast media administration make it possible to achieve MR imaging as not only morphological, but also functional and metabolic imaging tools for thoracic malignancies. Then, the Fleischner Society recommend MR imaging for lung diseases in 2020. In this review article, we focus on the MR imaging for lung cancer as well as pulmonary nodules and masses.

**Abstract:**

Since the Radiology Diagnostic Oncology Group (RDOG) report had been published in 1991, magnetic resonance (MR) imaging had limited clinical availability for thoracic malignancy, as well as pulmonary diseases. However, technical advancements in MR systems, such as sequence and reconstruction methods, and adjustments in the clinical protocol for gadolinium contrast media administration have provided fruitful results and validated the utility of MR imaging (MRI) for lung cancer evaluations. These techniques include: (1) contrast-enhanced MR angiography for T-factor evaluation, (2) short-time inversion recovery turbo spin-echo sequences as well as diffusion-weighted imaging (DWI) for N-factor assessment, and (3) whole-body MRI with and without DWI and with positron emission tomography fused with MRI for M-factor or TNM stage evaluation as well as for postoperative recurrence assessment of lung cancer or other thoracic tumors using 1.5 tesla (T) or 3T systems. According to these fruitful results, the Fleischner Society has changed its position to approve of MRI for lung or thoracic diseases. The purpose of this review is to analyze recent advances in lung MRI with a particular focus on lung cancer evaluation, clinical staging, and recurrence assessment evaluation.

## 1. Introduction

Since the introduction of magnetic resonance imaging (MRI) for the assessment of thoracic and lung diseases, various limitations of this procedure, mostly related to the relatively low proton density of lung parenchyma, the presence of cardiac and respiratory motion artifacts, and a long acquisition time, have hampered the clinical application of this technique. In 1991, the Radiology Diagnostic Oncology Group (RDOG) of the Radiological Society of North America (RSNA) reported on the accuracy of MRI, which had been used for unenhanced and non-electrocardiogram (ECG) -gated T1-weighted imaging (T1WI) and computed tomography (CT) for tumor classification and assessment of mediastinal node metastases. The accuracy of the two procedures was compared in a prospective cooperative study of 170 patients with non-small cell bronchogenic carcinoma (NSCLC) [1]. This study found no significant differences in sensitivity and specificity between T1WI and CT for distinguishing T3–T4 tumors from T0–T2 tumors and detecting mediastinal node metastases (N2 or N3), although T1WI had proven to be significantly more accurate than CT for the diagnosis of mediastinal invasion [1]. These findings resulted in very limited clinical applications of MRI in the 1990s.

During the first two decades of the 2000s, however, the introduction of technical advancements in MR systems, such as sequence and reconstruction methods, including parallel imaging techniques, and adjustments in the clinical protocol for gadolinium (Gd) contrast-media administration have resulted in frequent solutions for some of the limitations of these systems.

Furthermore, since 2000 various techniques have been developed that have validated the utility of MRI for lung cancer evaluations. These techniques include: (1) contrast-enhanced (CE-) MR angiography for T-factor evaluation, (2) short-time inversion recovery (STIR) turbo spin-echo (SE) sequences as well as diffusion-weighted imaging (DWI) for N-factor assessment, and (3) whole-body MRI with and without DWI and with positron emission tomography fused with MRI (PET/MRI) for M-factor or TNM stage evaluation, as well as for postoperative recurrence assessment of NSCLC, small cell lung cancer (SCLC) or other thoracic tumors using 1.5 tesla (T) or 3T systems [2,3,4,5,6,7,8,9,10]. Moreover, MRI for lung cancer is now covered by health insurance in North America, Eastern Asia, and Europe. In addition, the Fleischner Society has also changed its position on the approval of MRI for lung or thoracic diseases based on fruitful research results as well as other review articles [2,3,4,5,6,7,8,9,10,11,12,13,14,15,16]. The purpose of this review is to analyze recent advances in lung MRI with a particular focus on lung cancer evaluation, clinical staging, and recurrence assessment evaluation.

## 2. Dedicated Chest MRI

Lung cancer is the leading cause of tumor-related deaths worldwide [17,18]. To reduce the mortality rate for this disease, accurate staging is essential to determining the operability of and prognosis for patients so that clinicians can decide on the most suitable therapy. Since the early 1990s, the role of MR imaging has been found useful for solving a limited number of clinical issues. However, recent advancements in MRI have been demonstrated to be clinically beneficial for MRIs of patients with lung cancer. This section discusses the conventional and new MR techniques for TNM staging of lung cancer using dedicated chest MRI. Suggested protocols for dedicated chest MRI and whole-body MRI are listed and detailed in Table 1.

### 2.1. T-Factor Assessment

Many authors have used primary tumor (T) factor assessment to investigate the diagnostic capability of chest MRI for assessing the operability of patients with lung cancer (Table 2) [1,19,20,21,22,23]. At first, only conventional T1WI was tested, and no significant differences were observed between the accuracy of CT and MRI for the diagnosis of bronchial involvement or chest wall invasion, although T1WI was significantly more accurate than CT for the diagnosis of mediastinal invasion [1].

Unlike conventional MRI, only a few other techniques were tested in the late 1900s. Sakai et al. investigated the diagnostic accuracy of dynamic cine MRI for the detection of pleural invasion by assessing tumor movement through the parietal pleura during the respiratory cycle. Sensitivity, specificity, and accuracy of dynamic cine MRI were determined to be 100%, 70%, and 76%, respectively, while the corresponding values for conventional CT and MRI were 80%, 65%, and 68% [19]. Therefore, it was concluded at that time that dynamic cine MRI could be useful in detecting chest wall invasion.

In the early 2000s, CE-MR angiography was introduced as a promising method for the detection of cardiovascular or mediastinal invasion. One study assessed the capability of non-ECG-gated CE-MR angiography as compared with that of CT for pulmonary vasculature and left atrial invasion detection [20]. When CT was used in this study, invasion of the proximal pulmonary veins or the left atrium was suspected at 28 sites, obliteration of the pulmonary veins was identified at 9 sites, and the proximal portion of the pulmonary veins (within 1.5 cm from the left atrium) was judged to be involved at 19 sites. When MR angiography was used for these 28 sites, invasion of the left atrium was identified at 9 sites and of the pulmonary veins at 14 sites, whereas the pulmonary veins appeared normal in the remaining 5 sites. At the 14 sites with invasion, the distance of the pulmonary veins between the involved site and the entrance into the left atrium was ≥1.5 cm and <1.5 cm 7 sites each. Non-ECG-gated 3D CE-MR angiography is therefore suitable for assessing invasion of the pulmonary veins and the left atrium by lung cancer. Another study directly compared the diagnostic performance for mediastinal and vascular invasion detection among electrocardiogram (ECG)-gated T1WI, CE-MR angiography with and without ECG gating, and CT [21]. Sensitivity, specificity, and accuracy for detection of mediastinal and hilar invasion on ECG-gated CE-MR angiography ranged from 78% to 90%, 73% to 87%, and 75% to 88%, respectively, so that the diagnostic performance of ECG-gated CE-MR angiography was deemed superior to that of CT, T1WI, and conventional CE-MR angiography (Figure 1). Consequently, MRI became widely used for assessing mediastinal invasion until thin-section multiplanar reconstructed (MPR) images on multi-detector row CT (MDCT) became available [24].

In the 2010s, chest MRI systems started to change from using 1.5T to 3T MR systems, and we compared pulmonary vasculature anomaly evaluation for surgical procedure decisions for NSCLC patients among non-CE MR angiography, CE-MR angiography, and thin-section CE-CT [22]. For assessment of pulmonary arterial or venous anomalies, there were no significant differences in sensitivity, specificity, and accuracy among non-CE-MR angiography, CE-MR angiography, and thin-section CE-CT. In addition, interobserver agreement for each method was substantial or almost perfect. Therefore, results for the three methods were reproducible in this study. Furthermore, in a study comparing combined T1WI, T2WI, and 2D in-phase dynamic CE- T1-weighted gradient echo (GRE) sequences on a 3T MR system with MDCT for preoperative T-stage assessment for NSCLC, MRI was found to be slightly superior for more advanced tumors (T3, T4), especially for determining pleural and mediastinal invasion [23]. Despite the promising results of chest MRI for assessing the significance of the T-factor for lung cancer, the procedure remained in limited use, especially after MDCT had come into widespread use. Therefore, further technical developments, including the improvement of spatial and contrast resolutions, are warranted to render chest MRI suitable as a replacement for MDCT to improve T-factor evaluation for lung cancer patients.

### 2.2. N-Factor Assessment

The diagnostic performance of MRI for reginal lymph node (N) factor assessment has been considered similar to that of CT when diagnostic criteria consisting of measurements of the lymph node short-axis diameter are used to differentiate metastatic from non-metastatic lymph nodes. In addition, a few papers published in the 1990s suggested that the only advantage of MRI over CT was related to the multi-planar capability of MRI to detect lymph nodes in some areas, such as the aorto-pulmonary window and sub-carinal regions [1,25]. However, STIR imaging and DWI have been introduced as new and promising techniques for N-staging of NSCLC patients (Figure 2), and data for their diagnostic performances are shown in Table 3 [26,27,28,29,30,31,32,33,34,35,36].

Previous studies reported the sensitivity of quantitatively and qualitatively assessed STIR imaging, on a per-patient basis, ranging from 82.8% to 100.0%, from 89.2% to 93.1% for specificity, and from 82.8% to 92.2% for accuracy, and these values were equal to or higher than those for contrast enhanced computerized tomography (CE-CT) and F-fluorodeoxyglucose-positron emission tomography (FDG-PET) or PET fused with CT (PET/CT) [26,27,28,29,30,31,32,33,34,35,36]. On the other hand, the combination of FDG-PET/CT and STIR imaging was found to be significantly more effective than FDG-PET/CT alone for the differentiation of metastatic from non-metastatic lymph nodes on a per-node and per-patient basis [31]. By employing the basic features of the contrast mechanism of STIR imaging, this technique can add information based on the net difference between T1 and T2 relaxation times as well as on node size. However, major MR vendors use STIR pulse sequences with a longer echo time (TE) and echo train length (ETL) for fat-suppressed T2WI. Using these sequences results in STIR imaging being mainly T2-weighted and less sensitive to changes in relaxation time due to lymph node metastasis. On the other hand, some major studies have suggested STIR imaging can be useful for improving the diagnostic performance of lymph node metastasis in NSCLC when STIR pulse sequences with shorter TE and short ETL are used [26,27,28,34,36,37]. For STIR imaging, the inversion time is approximately 80–150 ms, during which time longitudinal magnetization for virtually all tissues is negative when a 90° pulse is applied, after which recovery begins for most tissues. After the second 90° pulse, the T1 contrast and the T2 contrast are additive; that is, increasing the T1 of a tissue increases the tissue’s relative signal intensity, as does increasing its T2. Some investigators have shown that there are significant differences between malignant and benign nodes in terms of their T1 and T2 relaxation times [38,39,40,41]. Because many pathologic lesions show an increase in both T1 and T2, the addition of these two types of contrast with the STIR sequence produces a higher net tissue contrast [42]. As a result, significant differences between lymph nodes with and without metastasis were observed in these studies [26,27,28,34,36,37]. It would therefore be better to use STIR imaging with a shorter TE and shorter ETL as T1WI for better diagnostic performance in this setting.

In addition to STIR imaging, it has also been suggested that DWI can be useful for the quantitative and qualitative diagnosis of lymph node metastasis in lung cancer patients [29,30,31,32,33,34,35,36]. Sensitivity, specificity, and accuracy of DWI reported by these studies ranged from 69.2% to 100%, 88% to 100%, and 71% to 94%, respectively, and these findings appear to be equal to or better than those for FDG-PET or PET/CT [29,30,31,32,33,34,35,36]. In addition, results for two meta-analyses were published in 2016 [43,44], and these studies reported a high diagnostic accuracy of MRI for staging hilar and mediastinal lymph nodes in NSCLC on both a per-patient and per-node basis. Moreover, Peerlings et al. reported that the pooled estimates of MRI for sensitivity, specificity, and DOR were 0.87 (95% confidence interval [CI]: 0.78, 0.92), 0.88 (95% CI: 0.77, 0.94), and 48.1 (95% CI: 23.4, 98.9) on a per-patient basis. On a per-node basis, the corresponding measurements were 0.88 (95% CI: 0.78, 0.94), 0.95 (95% CI: 0.87, 0.98), and 129.5 (95% CI: 49.3, 340.0) [44]. These values were superior to the sensitivities (on a per-patient basis: 72–76%, on a per-node basis: 61–78%) and specificities (on a per-patient basis: 88–91%, on a per-node basis: 90–95%) obtained with other meta-analyses of FDG-PET or PET/CT for the diagnosis of lymph node metastasis in SCLC [45,46,47,48]. Moreover, a meta-analysis directly compared DWI and FD-PET/CT and determined that both modalities are useful for detecting lymph node metastases in lung cancer without any significant differences between the two procedures. DWI might thus be an alternative modality for evaluating the nodal status of NSCLC [49]. Since the middle of the 2010s, there has been a change in MRI for N-staging from using 1.5T to 3T MR systems. However, an increased field strength is required for DWI to attain better image quality for evaluating smaller lymph nodes or for more accurately measuring the apparent diffusion coefficient (ADC) for the diagnosis of lymph node metastasis because susceptibility artifacts on a 3T MR system are more severe than those on a 1.5T MR system. Therefore, new techniques have been introduced to overcome this problem on DWI, such as DWIs obtained with fast advanced spin-echo rather than echo-planar imaging or DWIs computationally generated at the appropriate b value from actually obtained DWIs with different b values [36,37]. Both techniques have shown promising results, and their diagnostic performance is reportedly equal to that of STIR imaging and superior to that of FDG-PET/CT [36,37].

## 3. Whole-Body MRI and PET/MRI

Whole-body MRI and PET fused with MRI (PET/MRI) are new imaging methods for use with thoracic oncology patients and have been evaluated for their TNM stage assessment or recurrence evaluation of not only NSCLC but also other thoracic tumors, including thymic epithelial tumor, malignant mesothelioma, and small cell lung cancer [6,7,8,9,10,50,51,52,53,54,55,56,57,58,59,60,61,62,63,64,65,66,67]. Whole-body MR imaging has been introduced as another whole-body technique for the assessment of the TNM stage of lung cancer patients when ionizing radiation exposure is considered necessary. Moreover, this technique makes it possible to obtain information from various sequences with and without the administration of contrast media as a result of improvements in temporal resolution due to the use of new developments consisting of a parallel imaging technique, a moving table technique, and/or the use of multiple body-array coils. More recently, whole-body PET/MRI has been introduced as a new whole-body imaging technique consisting of FDG-PET combined with whole-body MRI for thoracic oncologic patients [56,57,58,59,60,61,62,63,64,65,66,67]. Two fundamentally different PET/MRI designs have been introduced. For the first of these designs, the PET and MRI data are acquired sequentially, either by obtaining sequential PET/CT and MR in the same space or by obtaining PET/CT and MR data in two different spaces or at different times. The main advantages of this design are the absence of an electromagnetic interference between the PET and MRI components, and a lack of need for extensive technical modifications of the individual systems. Additional advantages include the ability to use the PET/CT and MRI scanners independently for better use of resources, the capability to acquire MRI data during the uptake stage, the possibility of cost savings by upgrading the MRI and PET technologies independently, and the preservation of partial functionality if one of the components (PET or MRI) causes technical difficulties. Potential limitations of such systems include image misregistration due to patient motion (including differential bladder filling) and the need for a large installation space, while the impossibility of acquiring PET and MRI data simultaneously may also be a drawback. The second PET/MRI design features truly integrated systems. In this design, avalanche photodiodes such as lutetium oxyorthosilicate PET detectors are fitted in between the MR body and the gradient coils, or the more recently introduced semiconductor PET detectors (silicon-based photomultipliers) are used to create time-of-flight capabilities. With an integrated PET/MR system, multiple MR sequences are acquired while PET emission scan data are collected, so that imaging time is reduced and image misregistration is minimized. MRI-based photon attenuation correction can be attained with segmentation-based or atlas-based algorithms. However, this system makes it difficult to use Gd contrast media, which affects the attenuation correction of the PET section of a PET/MR examination. Therefore, the possibility of administering Gd contrast media as part of the imaging protocol for the MR section of the PET/MRI examination needs to be considered. Table 4 shows the recommended protocol for T, N, and M stage evaluation on whole-body MRI or the MR section of PET/MRI. Diagnostic performances of M-factor evaluation of lung cancer on whole-body MRI, as well as TNM stage assessments on whole-body PET/MRI, are also described in this section.

### 3.1. T-Factor Assessment

Evaluation of T-factor assessment using whole-body MRI of lung cancer started in 2008, and several reports have been published in recent years. In addition, whole-body PET/MRI started to be used in 2015 for assessing TNM stage evaluation by means of either co-registered or integrated PET/MRI. Table 5 shows data for the diagnostic performances of whole-body MRI or PET/MRI, as well as PET/CT for T-factor evaluation of lung cancer patients.

Several studies have compared the diagnostic performance for T-factor assessment of NSCLC by whole-body MRI, PET/MRI, or PET/CT with that of PET/CT or in terms of T-factor assessment [51,54,59,60,61,62,65]. These studies reported a diagnostic accuracy of whole-body MRI ranging from 63% to 94.3% of PET/MRI with and without signal intensity ranging from 65% to 94.3%, and of PET/CT ranging from 56% to 91.4%, with no significant differences observed among the three procedures (Figure 3). In addition to these studies, Schaarschmidt, et al. compared CE-PET/MRI and PET/CT and concluded that there was no impact on the management of patients, although some differences for T-factor evaluation were observed between these modalities [62]. It can therefore be presumed that there are no significant differences in diagnostic performance for T-factor evaluation among whole-body MRI, PET/MRI, and PET/CT when current state-of-the-art techniques are used for all three methods. Yet another study demonstrated whole-body MRI and PET/MRI used for not only morphological but also signal intensity change evaluations could attain better diagnostic performance for T-factor evaluation than could those two methods without signal intensity change evaluation or PET/CT [65]. Therefore, the use of morphological and relaxation time-based information for T-factor evaluation is recommended for whole-body MRI or PET/MRI in routine clinical practice.

### 3.2. N-Factor Assessment

The efficacy of whole-body MRI or PET/MRI for assessing the N-factor on both 1.5T and 3T MRI, with and without DWI, has been investigated [51,54,59,60,61,62,65]. N-factor assessment of lung cancer on whole-body MRI started in 2008, while the assessment of whole-body PET/MRI for TNM stage evaluation started in 2015. Table 6 shows data for the diagnostic performances of whole-body MRI or PET/MRI, as well as PET/CT for N-factor evaluation of lung cancer patients.

Several studies have compared the diagnostic performance for N-factor assessment of NSCLC by whole-body MRI, PET/MRI, or PET/CT with that of PET/CT [51,54,59,60,61,62,65]. These studies reported a diagnostic accuracy of whole-body MRI ranging from 66% to 98.6% of PET/MRI with and without signal intensity ranging from 57.1% to 98.6%, and of PET/CT ranging from 52.4% to 92.1%. One of these studies [59] used a 3T MR system for a direct comparison of N-factor evaluation by whole-body MRI, PET/MRI, and PET/CT for NSCLC patients and reported that the sensitivity and accuracy of whole-body MRI (sensitivity: 100%, accuracy: 98.6%) and PET/MRI with signal intensity assessment (sensitivity: 100%, accuracy: 98.6%) were significantly higher than those of PET/MRI without signal intensity assessment (sensitivity: 93.8%, accuracy: 92.1%) and PET/CT (sensitivity: 93.8%, accuracy: 92.1%), while there were no significant differences between the latter two. Another study [65] used 3T and 1.5T systems for the same patients for a direct comparison of N-factor evaluation by whole-body MRI, PET/MRI, and PET/CT, which demonstrated that the diagnostic accuracies of whole-body MRIs at 3T (86.5%) and 1.5T (84.6%) systems and PET/MRIs at 3T (84.6%) and 1.5T (82.7%) systems were significantly higher than that of PET/CT (79.8%), when lymph node metastasis of NSCLC was diagnosed by visual evaluation using relaxation time-based information. These studies therefore concluded that relaxation time-dependent information based on STIR imaging could improve the diagnostic performance for N-factor analysis on PET/MRI as well as whole-body MRI and that it would thus be better to use metabolic information for PET/MRI as well as PET/CT since the clinical relevance of whole-body MRI as well as PET/MRI was demonstrated in routine clinical practice (Figure 4).

### 3.3. M-Factor Assessment

CE-CT, bone scintigraphy, brain CE-MRI, and PET/CT are currently the most widely established and endorsed imaging modalities for the evaluation of distant metastases of lung cancer in routine clinical practice [68]. However, as a result of technical advances such as the introduction of fast imaging and moving table equipment, whole-body MRI has become a clinically feasible imaging modality and has been evaluated for M-factor or recurrence assessment since 2007 [50,51,52,53,54,55]. Table 7 shows data for the diagnostic performances of whole-body MRI or PET/MRI, as well as PET/CT for M-factor evaluation of lung cancer patients. Among the published studies comparing whole-body MRI and PET/CT for metastasis (M) factor assessment in lung cancer, the diagnostic capability of whole-body MRI without DWI (sensitivity: 60%–80%; specificity: 80%–96.4%; accuracy: 80%–94.8%) was found to be equal in some cases or was significantly different in others from that of FDG-PET/CT (sensitivity: 62.5%–97%; specificity: 94.5%–95.4%; accuracy: 73.3%–95.5%) [50,51,52,59]. On the other hand, the diagnostic performance of whole-body MRI with DWI was not significantly different from that of PET/CT, with sensitivity, specificity, and accuracy ranging from 70% to 100%, 80% to 92%, and 80% to 98.6%, respectively [50,51,52,59]. Furthermore, the diagnostic performance of whole-body DWI alone was rated as follows: sensitivity, 57.5%; specificity, 87.7%; and accuracy, 81.8%. In addition, the specificity and accuracy of whole-body DWI alone were significantly lower than those of whole-body MRI with and without DWI and PET/CT [52]. In contrast to the M-factor assessment, bone metastasis evaluation on the whole-body DWI showed significantly lower sensitivity and accuracy, and higher specificity than that on whole-body MRI without DWI and bone scan [53]. The same study found that the sensitivity, specificity, and accuracy of whole-body MRI with DWI were significantly higher than those of whole-body DWI, whole-body MRI without DWI, bone scan, or FDG-PET/CT in the same setting. Taking the above-mentioned results into account, a meta-analysis revealed that pooled estimates of sensitivity and specificity of PET/CT were 0.83 (95% confidence interval [CI], 0.54–0.95) and 0.93 (95% CI, 0.87–0.96), pooled sensitivity and specificity of whole-body MRI were 0.92 (95% CI, 0.18–1.00) and 0.93 (95% CI, 0.85–0.95), and pooled sensitivity and specificity for whole-body DWI were 0.78 (95% CI, 0.46–0.93) and 0.91 (95% CI, 0.79–0.96), respectively. The meta-analysis showed no statistical difference between the diagnostic odds ratio of whole-body MRI and DWI and that of PET/CT [69]. These findings therefore suggest that DWI should be used as a constituent of whole-body MR examination for improving diagnostic performance in routine clinical practice.

More recently, several investigators, who also evaluated whole-body PET/MRI in comparison with PET/CT or whole-body MRI for M-stage assessments of lung cancer patients reported that the diagnostic accuracy of PET/MR was in some cases equal to or in others higher than that of PET/CT, ranging from 81% to 98.6% for PET/MRI, from 83% to 98.7% for PET/CT, and from 94.2% to 98.6% for whole-body MRI [60,61,62,63,66] (Figure 5). More recently, the 3D GRE sequence using ultra-short TE less than 200 μs was introduced as a promising sequence for the whole-body MRI or PET/MRI. It was suggested to be useful for nodule detection and characterization, and to have equal or superior potential to that of a thin-section CT [70,71,72,73,74,75,76]. Despite the low number of published studies about M-factor assessment, whole-body MRI and whole-body PET/MR may well constitute feasible and promising imaging tools for M-factor evaluation of lung cancer patients in routine clinical practice, although further investigation of this potential function of the two procedures is clearly warranted.

## 4. Conclusions

To provide an update on the most recent advancements in the field of lung cancer TNM stage evaluation, we reviewed state-of-the-art MRI currently used for dedicated chest MRI, whole-body MRI, or PET/MRI. Following the implementation of fast imaging and the technical improvements of gradients and coil systems, MRI is now capable of providing unprecedented contrast with the aid of recently developed sequences, with and without contrast media, and using quantitative and qualitative analysis for thoracic oncology.

## Figures and Tables

**Figure 1 cancers-15-00950-f001:**
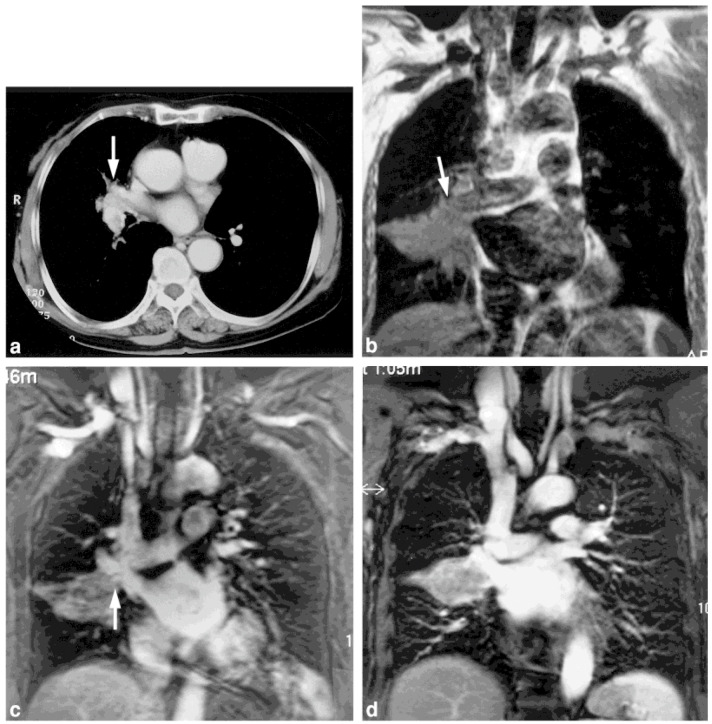
A 75-year-old man with squamous cell carcinoma in the middle lobe (permission obtained from reference #21). (**a**) Contrast-enhanced CT image demonstrates invasion of the truncus interlobar pulmonary artery and right superior pulmonary vein by the tumor (arrow). (**b**) ECG-gated T1-weighted SE image shows stenosis of the truncus interlobar pulmonary artery. Moreover, the flow-related enhancement within the right superior pulmonary vein (arrow) resembles invasion. (**c**) Conventional MRA image reveals irregularities in the wall of the superior pulmonary vein (arrow). Invasion of the right superior pulmonary vein by the tumor was also suspected. Cardiac motion artifacts have degraded the image quality. (**d**) ECG-triggered MRA image clearly shows the smooth vessel wall of the right superior pulmonary vein, with no invasion of this vein detected. Right bilobectomy was performed, and invasion of the superior pulmonary vein was not observed during surgery.

**Figure 2 cancers-15-00950-f002:**
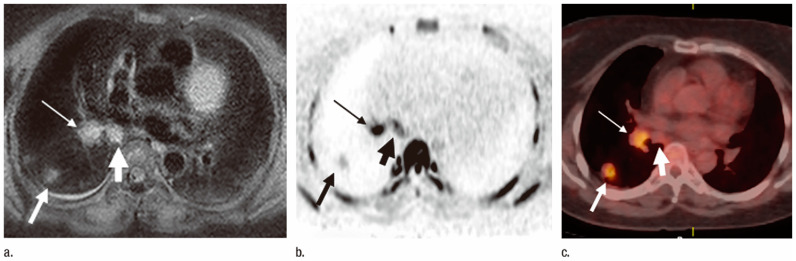
Images of a 73-year-old patient with pathologically diagnosed N2 adenocarcinoma (permission obtained from reference #34). (**a**) STIR turbo SE image shows primary lesion (medium-thick arrow), subcarinal lymph node (thick arrow), and right hilar lymph node (thin arrow) with high signal intensity (SI). The primary lesion in the right lower lobe is visible in the same axial plane. LSRs of lymph nodes were 0.75 (right hilar lymph node) and 0.78 (subcarina lymph node), LMRs were 1.7 (right hilar lymph node) and 1.9 (subcarina lymph node), and visual scores were 5. An accurate diagnosis of N2 disease was made. (**b**) DW MR image shows primary lesion (medium-thick arrow), subcarina lymph node (thick arrow), and right hilar lymph node (thin arrow) with high SI. The primary lesion in the right lower lobe is visible in the same axial plane. ADCs of lymph nodes were 2.8 × 10^−3^ s/mm^2^ (right hilar lymph node) and 3.4 × 10^−3^ s/mm^2^ (subcarina lymph node), and visual scores were 5. An accurate diagnosis of N2 disease was made. (**c**) FDG PET/CT image shows primary lesion (medium arrow) and right hilar lymph node (thin arrow) with high uptake of FDG and subcarina lymph node (thick arrow) with low uptake of FDG. Primary lesion in the right lower lobe is visible in the same axial plane. SUV_max_ of lymph nodes was 3.2 (right hilar lymph node) and 1.5 (subcarina lymph node), and visual scores were 5 (right hilar lymph node) and 2 (subcarina lymph node). An inaccurate diagnosis of N1 was made.

**Figure 3 cancers-15-00950-f003:**
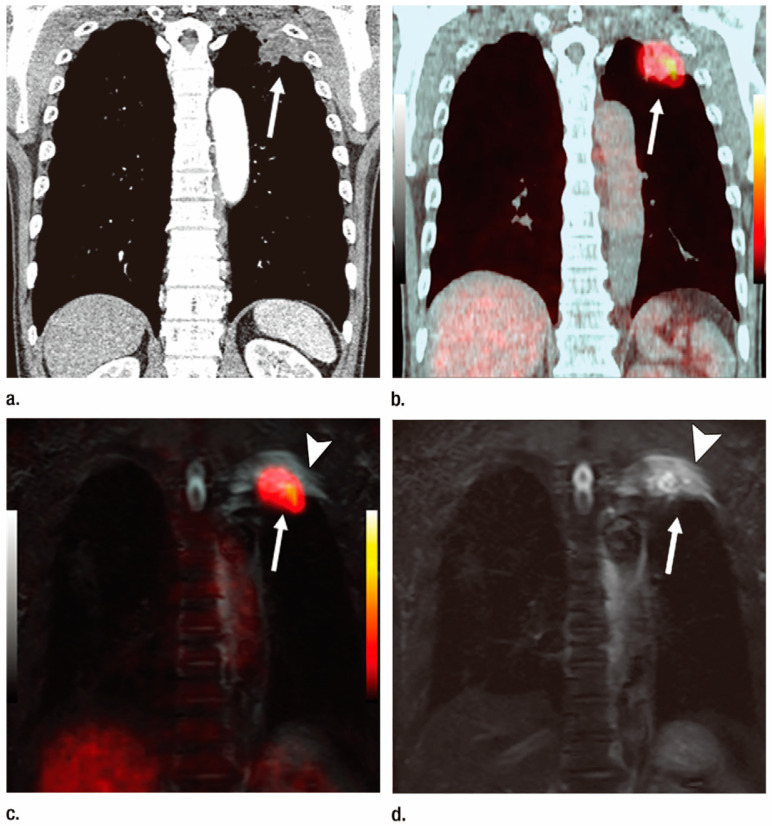
Squamous cell carcinoma diagnosed as T3N0M0 and assessed as stage IIB in a 57-year-old man (permission obtained from reference #59). (**a**) Contrast-enhanced thin-section multiplanar reformatted image with mediastinal window setting shows a mass (arrow) in the left upper lobe. Although the mass is attached to the chest wall, there were no osteoritic changes in the ribs or abnormal enhancement of the chest wall. This patient was diagnosed with T2a and inaccurately evaluated on contrast-enhanced CT imaging. (**b**) Integrated FDG PET/CT scan with mediastinal window setting shows a mass with high FDG uptake (arrow) in the left upper lobe. Although the mass was attached to the chest wall, there were no osteoritic changes in the ribs or abnormal FDG uptake in the chest wall. This patient was diagnosed as T2aN0M0 and stage IB and inaccurately evaluated on PET/CT imaging. (**c**) Co-registered PET/MRI image shows a mass with high FDG uptake (arrow) in the left upper lobe. Although there is FDG uptake within the mass, a high signal intensity area was observed in the adjacent chest wall (arrowhead), and the patient was diagnosed as T3. Assessment of the abnormal signal intensity within the chest wall, aided by whole-body PET/MRI with signal intensity assessment, led to a diagnosis of this patient as T3N0M0 and stage IIB. However, when this abnormal intensity was not included in the evaluation, this patient was diagnosed as T2aN0M0 and stage IB and thus could not be accurately diagnosed by using whole-body PET/MR without signal intensity assessment. (**d**) Whole-body MR images obtained with STIR FASE sequence show a mass (arrow) in the left upper lobe and high signal intensity area within the adjacent chest wall (arrowhead). Based on this abnormal intensity within the chest wall, this patient was accurately diagnosed on whole-body MR imaging as T3N0M0 and stage IIB.

**Figure 4 cancers-15-00950-f004:**
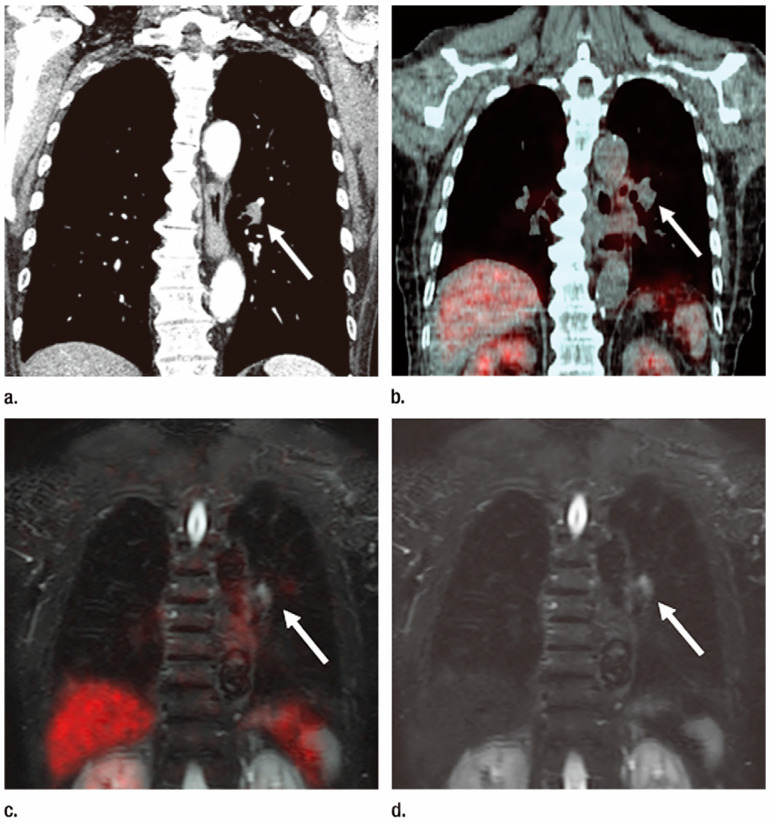
Invasive adenocarcinoma diagnosed as T1bN1M0 and evaluated as stage IIA in a 78-year-old man (permission obtained from reference #59). (**a**) Contrast-enhanced thin-section multiplanar reformatted image shows lymphadenopathy (arrow) with an 8-mm short-axis diameter in the left hilum. This lymph node was diagnosed as nonmetastatic with contrast-enhanced CT imaging; the patient was diagnosed inaccurately. (**b**) Integrated FDG PET/CT image with mediastinal window setting shows a lymph node without high FDG uptake (arrow). This patient was assessed with PET/CT imaging and was classified as T1bN0M0 and stage IA and thus not accurately evaluated with PET/CT imaging. (**c**) Co-registered PET/MR image, which combines FDG uptake with STIR FASE imaging, shows the left hilar lymph node (arrow) with high signal intensity (SI) and a minor amount of FDG uptake. When assessed with SI, this lymph node was diagnosed as metastatic. This patient was therefore diagnosed as T1bN1M0 and stage IIA and accurately assessed by using whole-body PET/MRI with SI assessment. However, when this abnormal SI was not included in the evaluation, whole-body PET/MRI without signal intensity assessment diagnosed this patient as T1bN0M0 and clinical stage IA. This patient could thus not be accurately evaluated by PET/MRI without signal intensity assessment. (**d**) Whole-body MR image obtained by using STIR FASE sequence displays the left hilar lymph node (arrow) with high signal intensity, and it was definitely diagnosed as metastatic. Therefore, this patient was diagnosed as T1bN1M0 and stage IIA and accurately evaluated by using whole-body MRI.

**Figure 5 cancers-15-00950-f005:**
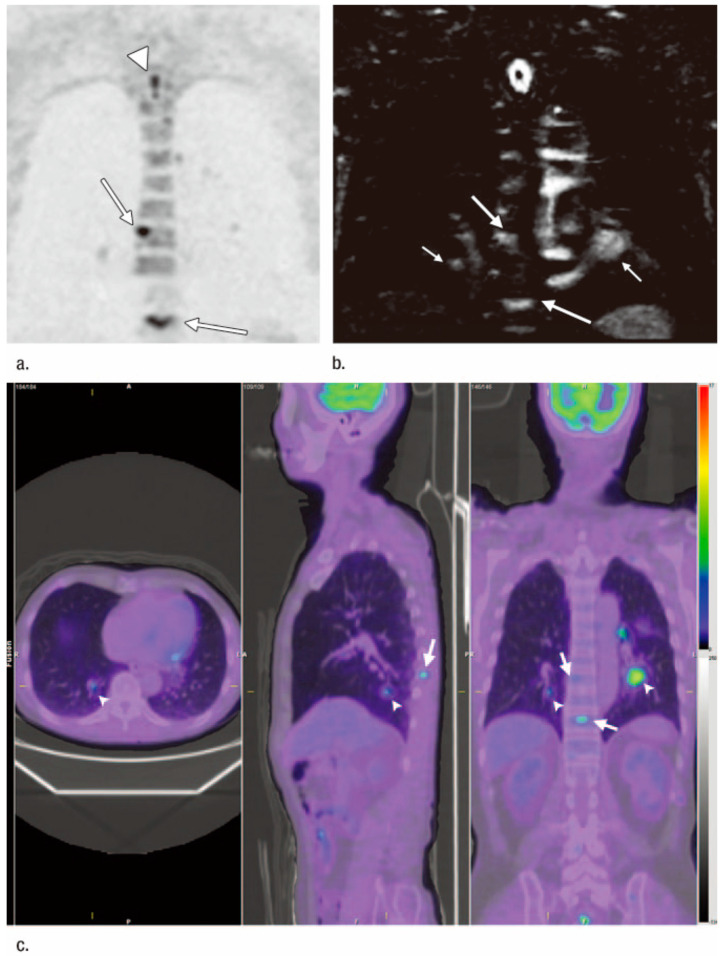
Images of 74-year-old man with adenocarcinoma, lung metastases, and bone metastases (permission obtained from reference #52). (**a**) Whole-body DW image (5759/70/180) in coronal plane shows bone metastases with a score of 5 for high signal intensity (arrows). However, the normal spinal cord also shows high signal intensity (arrowhead) and also received a score of 5. No lung metastases within either lung could be detected, and as these metastases were contained, they were scored as 1. Although this was a true-positive case, there were two false-negative sites and one false-positive site. (**b**) STIR image (3200/60/150) in coronal plane shows lung metastases (small arrows) and bone metastases (large arrows) as having high signal intensity, scored as 4 and 5, respectively. This was diagnosed as a true-positive case on whole-body MRI with and without DWI. (**c**) Integrated FDG-PET/CT images demonstrate bilateral lung metastases (arrowheads) and bone metastases (arrows), both of which were scored as 5. This was diagnosed as a true-positive case on integrated FDGPET/CT. Colored bar: standardized uptake value; gray bar: Hounsfield units.

**Table 1 cancers-15-00950-t001:** Recommended protocols for dedicated chest MRI.

	Sequence(s)	Comments
T-factor	Axial and coronal STIR FASE, FSE, or TSE	Useful for detecting mediastinal and/or thoracic wall invasion due to fat suppression
	Axial and coronal 3D T1-weighted GRE with and without Gd contrast media administration	Useful for assessing vascular invasion and measuring primary tumor
N-factor	Axial and coronal STIR FASE, FSE or TSE	High-accuracy detection and characterization of hilar and mediastinal lymph node metastasis
	Axial or coronal DWI using EPI or FASE (b = 0–1000 s/mm^2^)	
M-factor	Axial and/or coronal 3D GRE using UTE	Detection of contralateral (and ipsilateral) nodules
	Axial and coronal 3D T1-weighted GRE with and without Gd contrast media administration	Can detect pleural metastasis

STIR: short inversion-time (TI) inversion recovery (STIR), FASE: fast advanced spin-echo, FSE: fast spin-echo, TSE: turbo spin-echo, 3D: three-dimensional, GRE: gradient echo, Gd: gadolinium, DWI: diffusion-weighted imaging, EPI: echo-planar imaging, UTE: ultra-short echo time.

**Table 2 cancers-15-00950-t002:** Summary of diagnostic performance of T-factor assessment using dedicated chest MRI.

Author	Year	Field Strength	Imaging Method	Image Analysis	MRI			CT		
					SE (%)	SP (%)	AC (%)	SE (%)	SP (%)	AC (%)
Webb [1]	1991	0.35 and 1.5	Non-ECG-gated T1WI	Differentiation between T0–T2 and T3–T4	80	56	73	84	63	78
Sakai [19]	1997	1.5	Free-breathing cine gradient-echo (GRE)	Chest-wall invasion	10	70	76	80	65	68
Ohno [21]	2001	1.5	ECG-gated T1WI	Tumor invasion of pulmonary vessels	67–70	75–80	73–75	67–70	60–64	68–71
			Non-ECG-gated CE-MR angiography		78–80	73–83	75–82			
			ECG-gated MR angiography		89–90	83–87	86–88			
Ohno [22]	2014	3	Non-CE-MR angiography	Vascular anomaly in pulmonary artery or vein	50.0–77.1	97.4–98.5	87.7–93.2	62.5–91.4	88.9–100	90.4–95.9
			CE-MR angiography		62.5–77.1	97.4–100	87.7–95.9			
Tang [23]	2015	3	T1WI, T2WI with and without fat suppression, and 2D dynamic T1-weighted GRE	T-stage	N/A	N/A	82.2	N/A	N/A	84.4

ECG: electrocardiogram, GRE: gradient echo, 3D: three dimensional, SE: Sensitivity, SP: Specificity, AC: Accuracy, N/A: not applicable.

**Table 3 cancers-15-00950-t003:** Summary of diagnostic performance of N-factor assessment using dedicated chest MRI.

Author	Year	Field Strength (T)	Imaging Method	Reference Standard	Analysis	MRI			FDG-PET/CT			CT		
						SE (%)	SP (%)	AC (%)	SE (%)	SP (%)	AC (%)	SE (%)	SP (%)	AC (%)
Takenaka [26]	2002	1.5	STIR (T1-weighted)	Histology	Per node basis	100	96	96	N/A	N/A	N/A	52	91	83
			T1WI			52	91	83						
			STIR (T1-weighted)		Per patient basis	100	75	88				46	75	60
			T1WI			46	75	60						
Ohno [27]	2004	1.5	STIR (T1-weighted)	Histology	Per patient basis (Quantitative)	93	87	89	N/A	N/A	N/A	53	83	72
					Per patient basis (Qualitative)	88	86	86						
Ohno [28]	2007	1.5	STIR (T1-weighted)	Histology and/or follow-up	Per node basis (Quantitative)	89	99	98.2	82.3	96.2	65.9	N/A	N/A	N/A
					Per node basis (Qualitative)	86.3	97.2	96.3	80.8	95.8	94.6			
					Per patient basis (Quantitative)	90.1	93.1	92.2	76.7	87.5	83.5	N/A	N/A	N/A
					Per patient basis (Qualitative)	83.7	90.3	87.8	74.4	87.5	82.6			
Hasegawa [29]	2008	1.5	DWI	Histology	N2 vs. N0 or N1	80	97	95	N/A	N/A	N/A	N/A	N/A	N/A
Nomori [30]	2008	1.5	DWI	Histology and/or follow-up	Per node basis (Quantitative)	67	99	98	72	97	96	N/A	N/A	N/A
Morikawa [31]	2009	1.5	STIR (T2-weighted)	Histology	Per node basis (Quantitative)	96.3	67.3	84.7	90.2	65.5	80.3	N/A	N/A	N/A
					Per node basis (Qualitative)	93.9	70.9	84.7	86.6 (PET with qualitative STIR)	94.5 (PET with qualitative STIR)	89.8 (PET with qualitative STIR)	N/A	N/A	N/A
					Per patient basis (Quantitative)	N/A	N/A	N/A	90.2	65.6	81.7	N/A	N/A	N/A
					Per patient basis (Qualitative)	N/A	N/A	N/A	86.9 (PET with qualitative STIR)	96.9 (PET with qualitative STIR)	90.3 (PET with qualitative STIR)	N/A	N/A	N/A
Nakayama [32]	2010	1.5	STIR (T2-weighted)	Histology	Per patient basis (Quantitative)	61.5	98.1	91	N/A	N/A	N/A	N/A	N/A	N/A
			DWI		Per patient basis (Qualitative)	69.2	100	94						
Usuda [33]	2011	1.5	DWI	Histology	Per node basis (Quantitative)	75	99	95	48	97	90	N/A	N/A	N/A
					Per patient basis (Quantitative)	N/A	N/A	71	N/A	N/A	65	N/A	N/A	N/A
Ohno [34]	2011	1.5	STIR (T1-weighted)	Histology	Per node basis (Quantitative)	81.5 (LMR) or 83.7 (LSR)	85.9 (LMR) or 86.7 (LSR)	83.7 (LMR) or 85.1 (LSR)	75.6	88.8	82.2	N/A	N/A	N/A
			DWI		Per node basis (Quantitative)	74.8	87.4	81.1						
			STIR (T1-weighted)		Per node basis (Qualitative)	80	84.4	82.2	71.9 *	88.9	80.4	N/A	N/A	N/A
			DWI		Per node basis (Qualitative)	72.6 *	87.4	80						
			STIR (T1-weighted)		Per patient basis (Quantitative)	82.8 (LSR and LMR)	89.2 (LSR and LMR)	86.8 (LSR and LMR)	74.2	92.4	85.6	N/A	N/A	N/A
			DWI		Per patient basis (Quantitative)	74.2	90.4	84.4						
			STIR (T1-weighted)		Per patient basis (Qualitative)	77.4	88.5	84.4	69.9	91.7	83.6	N/A	N/A	N/A
			DWI		Per patient basis (Qualitative)	71	89.8	82.8						
Kim [35]	2012	1.5	Combined DWI, T2WI or PET/CT (Inclusive criteria)	Histology	Per node basis (Semi-quantitative)	69	93	89	46	96	87	N/A	N/A	N/A
			Combined DWI, T2WI and PET/CT (Exclusive criteria)		Per node basis (Semi-quantitative)	44	99	89						
			Combined DWI, T2WI or PET/CT (Inclusive criteria)		Per patient basis (Semi-quantitative)	N/A	N/A	71	N/A	N/A	63	N/A	N/A	N/A
			Combined DWI, T2WI and PET/CT (Exclusive criteria)		Per patient basis (Semi-quantitative)									
Ohno [36]	2015	3	STIR (T1-weighted)	Histology	Per node basis (Qualitative)	82.1	98.7	90.4	57.7	97.4	77.6	N/A	N/A	N/A
			DWI obtained by FASE sequence		Per node basis (Qualitative)	82.1	98.7	90.4						
			DWI obtained by EPI sequence		Per node basis (Qualitative)	60.3	98.7	79.5						
			STIR (T1-weighted)		Operative vs. Inoperative (Qualitative)	100	88	89.5	50	89.2	84.2	N/A	N/A	N/A
			DWI obtained by FASE sequence		Operative vs. Inoperative (Qualitative)	100	88	89.5						
			DWI obtained by EPI sequence		Operative vs. Inoperative (Qualitative)	75	89.2	87.4						
Ohno [37]	2022	3	STIR (T1-weighted)	Histology	Per node basis (Quantitative)	86.8	66.7	76.8	78.9	71.1	75	N/A	N/A	N/A
			Actual DWI		Per node basis (Quantitative)	83.3	66.7	75						
			ADC map from actual DWI		Per node basis (Quantitative)	81	71.9	76.8						
			Computed DWI at the most appropriate b value		Per node basis (Quantitative)	86.8	71.9	79.4						
			STIR (T1-weighted)		Per node basis (Qualitative)	86.8	60.5	73.7	85.1	57	71	N/A	N/A	N/A
			Actual DWI		Per node basis (Qualitative)	82.5	60.5	71.5						
			ADC map from actual DWI		Per node basis (Qualitative)	82.5	60.5	71.5						
			Computed DWI at the most appropriate b value		Per node basis (Qualitative)	87.7	60.5	74.1						
			STIR (T1-weighted)		Per patient basis (Quantitative)	N/A	N/A	90.6	N/A	N/A	85.3	N/A	N/A	N/A
			Actual DWI		Per patient basis (Quantitative)			86.9						
			ADC map from actual DWI		Per patient basis (Quantitative)			86.9						
			Computed DWI at the most appropriate b value		Per patient basis (Quantitative)			90.2						
			STIR (T1-weighted)		Per patient basis (Qualitative)	N/A	N/A	86.9	N/A	N/A	82.9	N/A	N/A	N/A
			Actual DWI		Per patient basis (Qualitative)			84.5						
			ADC map from actual DWI		Per patient basis (Qualitative)			84.5						
			Computed DWI at the most appropriate b value		Per patient basis (Qualitative)			86.5						

*: SE: Sensitivity, SP: Specificity, AC: Accuracy, STIR: short inversion-time (TI) inversion recovery (STIR), DWI: diffusion-weighted imaging, LSR: liquid silicone rubber, LMR: lossy mode resonance, ADC: apparent diffusion coefficient (ADC), N/A: not applicable.

**Table 4 cancers-15-00950-t004:** Recommended protocols for whole-body MRI or PET/MRI.

	Whole-Body MR Imaging or MR Section of Whole-Body PET/MRI	
	Sequences	Comments
T-factor	Coronal or axial STIR imaging	Can detect mediastinal and/or thoracic wall invasion due to fat suppression
	Coronal or axial T2WI	
	Coronal 3D T1-weighted GRE with or without Gd contrast media administration	Useful for assessing vascular invasion
N-factor	Coronal or axial STIR imaging	High accuracy for detection and characterization of hilar and mediastinal lymph node metastasis
	Coronal or axial DWI using EPI or FASE (b = 0–1000 s/mm^2^)	
M-factor	Coronal, sagittal, or axial STIR imaging	Detection of distant metastases (e.g., cerebral, adrenal, skeletal, abdominal, or lymph nodes)
	Coronal, sagittal, or axial T1-weighted GRE in-phase/out-phase	
	Coronal or axial 3D GRE using UTE	
	Coronal DWI or axial using EPI or FASE (b = 0–1000 s/mm^2^)	
	Coronal, sagital, or axial 3D T1-weighted GRE with or without Gd contrast media administration	

STIR: short inversion-time (TI) inversion recovery (STIR), FASE: fast advanced spin-echo, FSE: fast spin-echo, TSE: turbo spin-echo, 3D: three dimensional, GRE: gradient echo, Gd: gadolinium, DWI: diffusion-weighted imaging, EPI: echo-planar imaging, UTE: ultra-short echo time.

**Table 5 cancers-15-00950-t005:** Summary of diagnostic performance of T-factor assessment using whole-body MRI or PET/MRI.

Author	Year	Field Strength (T)	Image Evaluation	Whole-Body MRI			FDG-PET/MRI				FDG-PET/CT		
				SE (%)	SP (%)	AC (%)	PET/MR Method	SE (%)	SP (%)	AC (%)	SE (%)	SP (%)	AC (%)
Yi [51]	2008	3	Visual assessment	N/A	N/A	86	N/A	N/A	N/A	N/A	N/A	N/A	82
Sommer [54]	2012	1.5	Visual assessment	N/A	N/A	63	N/A	N/A	N/A	N/A	N/A	N/A	56
Ohno [59]	2015	3	Visual assessment with signal intensity	100	55.6	94.3	Co-registered	100	55.6	94.3	100	33.3	91.4
			Visual assessment without signal intensity					100	33	91.4			
Huellner [60]	2016	3	Visual assessment	N/A	N/A	N/A	Integrated	N/A	N/A	69	N/A	N/A	81
Lee [61]	2016	3	Visual assessment	N/A	N/A	N/A	Integrated	N/A	N/A	80	N/A	N/A	80
Schaarschmidt [62]	2017	3	Visual assessment	N/A	N/A	N/A	Integrated	N/A	N/A	65	N/A	N/A	65
Ohno [65]	2020	3	Visual assessment with signal intensity	N/A	N/A	92.3	Co-registered	N/A	N/A	92.3	N/A	N/A	94.2
		1.5	Visual assessment with signal intensity			92.3				89.4			

SE: Sensitivity, SP: Specificity, AC: Accuracy, N/A: not applicable.

**Table 6 cancers-15-00950-t006:** Summary of diagnostic performance of N-factor assessment using whole-body MRI or PET/MRI.

Author	Year	Field Strength (T)	Image Evaluation	Whole-Body MRI			FDG-PET/MRI				FDG-PET/CT		
				SE (%)	SP (%)	AC (%)	PET/MR Method	SE (%)	SP (%)	AC (%)	SE (%)	SP (%)	AC (%)
Yi [51]	2008	3	Visual assessment	N/A	N/A	68	N/A	N/A	N/A	N/A	N/A	N/A	70
Sommer [54]	2012	1.5	Visual assessment	N/A	N/A	66	N/A	N/A	N/A	N/A	N/A	N/A	71
Ohno [59]	2015	3	Visual assessment with signal intensity	100	92.9	98.6	Co-registered	100	92.9	98.6	93.8	85.7	92.1
			Visual assment without signal intensity					93.8	85.7	92.1			
Huellner [60]	2016	3	Visual assessment	N/A	N/A	N/A	Integrated	N/A	N/A	79	N/A	N/A	88
Lee [61]	2016	3	Visual assessment	N/A	N/A	N/A	Integrated	N/A	N/A	57.1	N/A	N/A	52.4
Schaarschmidt [62]	2017	3	Visual assessment	N/A	N/A	N/A	Integrated	N/A	N/A	77	N/A	N/A	77
Ohno [65]	2020	3	Visual assessment with signal intensity	N/A	N/A	86.5	Co-registered	N/A	N/A	84.6	N/A	N/A	79.8
		1.5	Visual assessment with signal intensity			84.6				82.7			

SE: Sensitivity, SP: Specificity, AC: Accuracy, N/A: not applicable.

**Table 7 cancers-15-00950-t007:** Summary of diagnostic performance of M-factor assessment using whole-body MRI or PET/MRI.

Author	Year	Field Strength (T)	Evaluated Sites	Image Evaluation	Whole-Body MRI				FDG-PET/MRI				FDG-PET/CT		
					Protocol	SE (%)	SP (%)	AC (%)	PET/MR Method	SE (%)	SP (%)	AC (%)	SE (%)	SP (%)	AC (%)
Ohno [50]	2007	1.5	M-factor	Visual assessment	MRI including brain MRI	80	80	80	N/A	N/A	N/A	N/A	70	74.3	73.3
					MRI excluding brain MRI	80	80	80					80	74.3	75.6
Yi [51]	2008	3	M-factor	Visual assessment	N/A	52	94	86	N/A	N/A	N/A	N/A	48	96	86
Ohno [52]	2008	1.5	M-factor	Visual assessment	DWI	57.5	87.7	81.8	N/A	N/A	N/A	N/A	62.5	94.5	88.2
			M-factor		MRI without DWI	60	92	85.7							
			M-factor		MRI with DWI	70	92	87.7							
Takenaka [53]	2009	1.5	bone metastasis	Visual assessment	DWI	95.5	93.7	93.9	N/A	N/A	N/A	N/A	97	95.4	95.5
					MRI without DWI	73.1	96.4	94.8							
					MRI with DWI	95.5	96.1	96.1							
					Bone scan	95.5	95.4	95.5							
Ohno [59]	2015	3	M-factor	Visual assessment with signal intensity assessment	MRI including DWI and brain MRI	100	87.5	98.6	Co-registered	100	87.5	98.6	92.7	75	90.7
				Visual assessment without signal intensity assessment						92.7	81.3	91.4			
Huellner [60]	2016	3	M-factor	Visual assessment	N/A	N/A	N/A	N/A	Integrated	N/A	N/A	81	N/A	N/A	83
Lee [61]	2016	3	M-factor	Visual assessment	N/A	N/A	N/A	N/A	Integrated	N/A	N/A	83.3	N/A	N/A	83.3
Schaarschmidt [62]	2017	3	M-factor	Visual assessment	N/A	N/A	N/A	N/A	Integrated	N/A	N/A	98.7	N/A	N/A	98.7
Ohno [65]	2020	3	M-factor	Visual assessment with signal intensity	MRI including DWI and brain MRI	N/A	N/A	97.1	Co-registered	N/A	N/A	97.1	N/A	N/A	96.2
		1.5		Visual assessment with signal intensity				94.2				94.2			

SE: Sensitivity, SP: Specificity, AC: Accuracy, N/A: not applicable.

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
