# Peer review of "State of the Art MR Imaging for Lung Cancer TNM Stage Evaluation"

_cancers, 2023, doi:10.3390/cancers15030950_

Round 1
Reviewer 1 Report
This review article well introduces the recent advances in lung MRI, focusing on its application to lung cancer evaluation, especially each T-, N-, M- staging and recurrence evaluation. It will be of interest to radiologists.
I would like to advise to add some minor revisions
Table 1
FASE needs to be spelled out at the foot notes.
i. T-factor assessment
As for the T-staging, descriptions about breathing dynamic MRI or cine MRI for the evaluation of direct invasion of lung cancer had better to be added. If the lung cancer moves separately from chest wall, mediastinum, heart, great vessels on breathing dynamic MRI, we can deny the invasion by the tumor.
Author Response
Reviewer #1’s comment and response:
I would like to advise to add some minor revisions.
Thank you for your comments.
- Table 1: FASE needs to be spelled out at the foot notes.
Although reviewer #1 asked to spell out "FASE" in the foot note, FASE had been already spelled out in Page 27.
- T-factor assessment
As for the T-staging, descriptions about breathing dynamic MRI or cine MRI for the evaluation of direct invasion of lung cancer had better to be added. If the lung cancer moves separately from chest wall, mediastinum, heart, great vessels on breathing dynamic MRI, we can deny the invasion by the tumor.
Although reviewer #1 suggest to be added the results of breathing dynamic MRI or cine MRI for evaluation of direct invasion of lung cancer, the results had already been stated in Page 5, line 1-7.

Reviewer 2 Report
Dear Doctor Ohno,
I enjoyed reading your long manuscript.
I do have a few comments / suggestions that you may use as you like:
Typos / syntax / style:
· Abstract: I would try to avoid acronyms
· Frequent solutions for of some of the limitations of these systems ïƒ typo / repetition
· In view of these changes, the Fleischner Society has changed its position to approval of MRI for lung or thoracic diseases ïƒ syntax
· This article will review these recent advances in lung MRI, focusing on its application to lung cancer evaluation, especially regarding lung cancer stage and recurrence evaluation ïƒ you do not need the future tense. Also, try to avoid repetitions. Try to simply say: “The purpose of this review is to analyze recent advances in lung MRI with a particular focus on lung cancer evaluation, clinical staging and recurrence assessment”
· Lines 57 – 69: you simply copy pasted the abstract. I would rather try to elaborate differently, guiding the reader thru the paper. Otherwise, again, your risk is losing the readers’ attention.
· You must adjust the column width on your tables to make them better readable.
· Pleural invasion by assessing tumor movement through the partial pleura during the ïƒ you probably meant ‘parietal’
· Lines 106 – 114: here you have an example of a lot of data over a lot of words. It shows you did a lot of work, but it may be counterproductive. I would rather summarize it all in max 3 sentences. Get to the point and convey a clear message.
· Figure 3 and table 7 are somehow misplaced towards the end of the paper.
Conclusion: in my opinion, it is a bit redundant and rather weak. I would try to focus on the true key aspects that you want the reader to remember, and to highlight what we need to do today to achieve your vision. For the same reason, I would also avoid standard sentences i.e. further studies are needed to confirm bla bla bla. Take the chance that you may publish this paper to truly make an impact.
Text: overall, even single sentences are too long (let alone groups of sentences / paragraphs). Your risk is losing the reader. It is a quite long review with a fair amount of data, including big tables. Try to shorten each sentence and to work on the content of each table.
30+ self-citations: I recognise the role of the first author as a KOL in this field, but I feel as though that is a inappropriately high number. Perhaps, I would rather enjoy seeing him championing a prospective large trial to make MR the new standard of care.
Author Response
Reviewer #2’s comments and answers:
I enjoyed reading your long manuscript.
Thank you for your comments.
I do have a few comments / suggestions that you may use as you like:
Thank you for your comments and suggestions. We revise this manuscript basically with reviewer #1 and #2’s comments, although some comments were not applied. Please check the revision.
Typos / syntax / style:
- Abstract: I would try to avoid acronyms
According to reviewer #2's comment #1, the abstract is modified as suggested.
- Frequent solutions for of some of the limitations of these systems typo / repetition
According to reviewer #2's comment #2, the miss typo is corrected in Page 2, line 5.
- In view of these changes, the Fleischner Society has changed its position to approval of MRI for lung or thoracic diseases syntax
According to reviewer #2's comment #3, the statement is modified in Page 2, line 14-16.
- This article will review these recent advances in lung MRI, focusing on its application to lung cancer evaluation, especially regarding lung cancer stage and recurrence evaluation you do not need the future tense. Also, try to avoid repetitions. Try to simply say: “The purpose of this review is to analyze recent advances in lung MRI with a particular focus on lung cancer evaluation, clinical staging and recurrence assessment”.
According to reviewer #2's comment #4, the statement is modified in Page 2, line 16-17 as suggested.
- Lines 57 – 69: you simply copy pasted the abstract. I would rather try to elaborate differently, guiding the reader thru the paper. Otherwise, again, your risk is losing the readers’ attention.
According to reviewer #2's comment #5, these statements are modified as compared with the Abstract. However, abstract is sometimes similar to Introduction. Therefore, we could not totally modify.
- You must adjust the column width on your tables to make them better readable.
Although we try o adjust the column width of Table, it is difficult to limited page width.
- Pleural invasion by assessing tumor movement through the partial pleura during the ïƒ you probably meant ‘parietal’
As reviewer #2's suggestion #7, miss typo is modified in Page 5, line 3. .
- Lines 106 – 114: here you have an example of a lot of data over a lot of words. It shows you did a lot of work, but it may be counterproductive. I would rather summarize it all in max 3 sentences. Get to the point and convey a clear message.
According to reviewer #2's comment #8, the statement is modified in Page 6, line 5-8.
- Figure 3 and table 7 are somehow misplaced towards the end of the paper.
In the original main text file, Figure 3 and Table 7 are not misplaced. Please check them in the originally submitted file.
- Conclusion: in my opinion, it is a bit redundant and rather weak. I would try to focus on the true key aspects that you want the reader to remember, and to highlight what we need to do today to achieve your vision. For the same reason, I would also avoid standard sentences i.e. further studies are needed to confirm bla bla bla. Take the chance that you may publish this paper to truly make an impact.
Thank you for your comments. This review is covered many parts of MRI for lung cancer. Therefore, some parts may be considered as redundant. However, in this time, we hope to publish in this version. If we correct, it takes longer time for preparation.
- Text: overall, even single sentences are too long (let alone groups of sentences / paragraphs). Your risk is losing the reader. It is a quite long review with a fair amount of data, including big tables. Try to shorten each sentence and to work on the content of each table.
Thank you for your comments. This paper had been checked native speaker who edited over 100 my original papers. Therefore, we hope to publish in this version.
- 30+ self-citations: I recognise the role of the first author as a KOL in this field, but I feel as though that is a inappropriately high number. Perhaps, I would rather enjoy seeing him championing a prospective large trial to make MR the new standard of care.
Thank you for your comments. However, almost all major papers about “MRI for lung cancer” had been published by us since 2000. In addition, current major review articles in this topic had been published by us since 2000. Therefore, self-citation can’t be avoid, when prepare review article about “MRI for lung cancer”.

Round 2
Reviewer 2 Report
Dear Authors,
I may be wrong, but it sounds like you are in a rush to publish your paper AS IS.
I was hoping you would use the first round of review to take the ball and run with it. Instead, it seems you brush our key points off. Moreover, you use general statements to justify your conduct:
1) “abstract is sometimes similar to Introduction. Therefore, we could not totally modify”
2) “if we correct, it takes longer time for preparation”
3) “This paper had been checked native speaker who edited over 100 my original papers” – “self-citation can’t be avoid”.
4) Etc. etc.
I am sorry, but I do not think this is the right attitude.
Best of luck!
Author Response
Reviewer #2’s comments and answers:
Dear Authors,
I may be wrong, but it sounds like you are in a rush to publish your paper AS IS.
I was hoping you would use the first round of review to take the ball and run with it. Instead, it seems you brush our key points off. Moreover, you use general statements to justify your conduct:
1) “abstract is sometimes similar to Introduction. Therefore, we could not totally modify”
According to reviewer #2’s comment 1), we modify the abstract as suggested.
2) “if we correct, it takes longer time for preparation”
According to reviewer #2’s comment 2), we modify the conclusion as suggested.
3) “This paper had been checked native speaker who edited over 100 my original papers” – “self-citation can’t be avoid”.
Thank you for your comments. This paper had been checked native speaker who edited over 100 my original papers. In addition, he briefly checks this version.
Although reviewer #2 suggest to avoid self-citations, almost all high quality major papers about “MRI for lung cancer”, which had been cited many papers, are our papers and can’t be avoided, when prepared a review paper in this topic. Therefore, we can’t follow reviewer #2’s comments. In addition, this paper is invited review article. So, no further corrections or revision are hard for us. Please understand this fact.